# Oxygen-Bonding State and Oxygen-Reduction Reaction Mechanism of $Pr_{0.7}Ca_{0.3}Mn_{1-x}Co_xO_{3-d}$ (x = 0, 0.1, 0.2, 0.3)

Kanghee Jo, Seungjae Lee  and Heesoo Lee *

School of Materials Science and Engineering, Pusan National University, Busan 46241, Republic of Korea; jokanghee@pusan.ac.kr (K.J.); bluejae2@pusan.ac.kr (S.L.)
* Correspondence: heesoo@pusan.ac.kr; Tel.: +82-051-510-2388

**Abstract:** We investigated the effects of Co doping on $Pr_{0.7}Ca_{0.3}MnO_{3-d}$ in the perspective of an oxygen-bonding state change. In all compositions, $Pr_{0.7}Ca_{0.3}Mn_{1-x}Co_xO_{3-d}$ (PCMCx, x = 0, 0.1, 0.2, 0.3) showed an orthorhombic structure, and the lattice gradually contracted with increasing Co content. The doped Co was mostly present as 2+ and 3+, which decreased the average oxidation value of the B site and created oxygen vacancies for charge compensation. However, as the Co content increased, the proportion of $Co^{3+}$ increased, and the content of oxygen vacancies gradually decreased. In addition, the ratio of adsorbed oxygen in PCMC0.1 was the highest, and the B-O covalency was enhanced. Accordingly, the electrochemical reaction of oxygen with the cathode material in PCMC0.1 could occur most easily, showing the smallest polarization resistance among the Co-doped $Pr_{0.7}Ca_{0.3}MnO_{3-d}$. We can confirm the formation of oxygen vacancies via Co doping and the effect of B-O covalency on the oxygen-reduction reaction of $Pr_{0.7}Ca_{0.3}MnO_{3-d}$.

**Keywords:** co-doped $Pr_{0.7}Ca_{0.3}MnO_{3-d}$; oxygen vacancy; B-O bond covalency; distribution of relaxation time (DRT); ORR mechanism

## 1. Introduction

A solid oxide fuel cell (SOFC) is an electrochemical cell that directly converts the chemical energy of fuel into electricity with an efficiency of up to 80% at high temperatures of 800~1000 °C using ceramic materials and is attracting attention as a tool for realizing carbon neutrality. High-temperature operating conditions cause high costs, deterioration of surrounding materials, and thermal shock during 'start–stop' cycles. Research to reduce the operating temperature to an intermediate temperature range (600~800 °C) is conducted to overcome the disadvantages [1–3]. To reduce the operating temperature, it is important to improve the sluggish oxygen-reduction reaction that occurs on the surface of the cathode. To improve it, it is important to control the material properties so that the reaction that mainly occurs at the air–electrode–electrolyte interface (triple-phase boundary, TPB) can occur at the electrode–air interface [4,5].

Co-based perovskite oxides such as $La_{0.8}Sr_{0.2}Co_{0.8}Fe_{0.2}O_{3-\delta}$ (LSCF) and $Ba_{0.5}Sr_{0.5}Co_{0.8}Fe_{0.2}O_{3-\delta}$ (BSCF) have been widely studied because of their ability to conduct oxygen ion and electrons simultaneously and high oxygen-reduction reaction (ORR) activity to reduce the operating temperature [6,7]. Double perovskite oxides such as $PrBaCo_2O_{5+d}$, $PrBaMn_2O_{5+d}$, and $NdBaMnO_{5+d}$, which show better activity by forming an oxygen pathway through cation ordering, have been recently studied [8–11]. However, Co-based perovskite oxides and double perovskite oxides have a large thermal expansion coefficient compared to ceria-based electrolytes, which can cause delamination of the electrode–electrolyte interface. Cobalt-free cathode materials such as $Ba_{0.5}Sr_{0.5}FeO_{3-d}$ are being studied [12]. In the case of $PrMnO_3$, Ishihara et al. showed that among various $Ln_{0.6}Sr_{0.4}MnO_3$ (Ln = La, Pr, Nd, Sm, etc.)-based perovskite oxides, $Pr_{0.6}Sr_{0.4}MnO_{3-d}$ exhibited the best electrical conductivity and, unlike the widely studied LSM, did not react

with the electrolyte $Y_2O_3$-stabilized $ZrO_2$, making it suitable as a cathode material for low- and intermediate-temperature SOFCs [13]. However, further research is needed to improve the electrode performance for a cobalt-free cathode. Small amounts of Co doping into a cobalt-free cathode have been studied [14].

We investigated the effect of Co doping on the B-O bonding structure and electrochemical properties in $Pr_{0.7}Ca_{0.3}MnO_{3-d}$. We synthesized $Pr_{0.7}Ca_{0.3}Mn_{1-x}Co_xO_{3-d}$ with different Co contents using an EDTA (ethylenediaminetetraacetic acid)-citric acid complex process, and the crystal structures were confirmed using X-ray diffraction patterns. Core level spectra were observed using X-ray photoelectron spectroscopy, and peak deconvolution was performed to confirm the oxidation and oxygen binding states of the metal ions. EIS was measured at 550~750 °C to confirm the electrochemical properties depending on the Co content, and changes in the ORR mechanism were analyzed using DRT analysis.

## 2. Materials and Methods

PCMCx ($Pr_{0.7}Ca_{0.3}Mn_{1-x}Co_xO_{3-d}$, x = 0, 0.1, 0.2, 0.3) powders with various compositions were synthesized using the EDTA-citrate complexing process (ECCP). $Pr(NO_3)_3 \cdot 6H_2O$ (99.9%, Sigma-Aldrich, St. Louis, MO, USA), $Ca(NO_3)_2 \cdot 4H_2O$ (98%, Sigma-Aldrich), $Co(NO_3)_3 \cdot 6H_2O$ (98%, Sigma-Aldrich), and $Mn(NO_3)_2 \cdot 4H_2O$ (98%, Sigma-Aldrich) metal precursors were dissolved in deionized water. Ethylenediaminetetraacetic acid (EDTA, 99.5%, Alfa Aesar, Haverhill, MA, USA) was added to a 1N $NH_4OH$ (Junsei chemical Co., Tokyo, Japan) solution to obtain a $NH_3$-EDTA buffer solution. The $NH_3$-EDTA and crystallized citric acid monohydrate (99.5%, Samchun Chemical, Pyeongtaek, Korea) powders, as chelating agents, were applied to the mixed-metal precursor solution to make a sol for a total metal ion: EDTA: citric acid molar ratio of 1:1:2. The sol was heated with stirring to evaporate the solvent. Clear gels were obtained, and then these gels were pre-calcined at 200 °C. Then, the pre-calcined precursors were calcined in air at 950 °C.

Symmetric cells (PCMCx | SDC | PCMCx) were prepared to investigate the electrochemical properties of the PCMCx powders. The SDC pellets were sintered at 1400 °C for 4 h using Sm-doped ceria powder (SDC20-HP, Fuelcellmaterials, Lewis Center, OH, USA). The PCMCx powders were mixed with a binder prepared from $\alpha$-terpineol and ethyl-cellulose to form PCMCx pastes, which were screen-printed onto both sides of the SDC pellets. After drying, the symmetric cells were calcined at 950 °C for 2 h in air.

To examine the crystal structures of the calcined and sintered powders, powder X-ray diffraction (XRD, PANalytical X'pert-Pro MPD PW3040/60) was performed at room temperature using a step-scan procedure ($0.02°/2\theta$ step, time per step 0.5 s) in the $2\theta$ range of 10–90°. X-ray photoelectron spectroscopy (XPS) was carried out to measure the core level spectra of the PCMCx powders using the K-ALPAH+ System (HPXPS, Al $K\alpha$ X-ray source (1486.6 eV), ThermoFisher Scientific, Waltham, MA, USA) at the Korea Basic Science Institute (KBSI), Busan, center. Impedance measurements were conducted using an IviumStat instrument (Ivium, Eindhoven, the Netherlands) over the frequency range of $10^6$–0.01 Hz with an excitation voltage of 10 mV at an operating temperature of 600–700 °C under open-circuit conditions in air. The electrochemical impedance spectroscopy (EIS) results were multiplied by 0.5 to account for the two electrodes. The impedance spectra data were further fitted using EC-lab software (4.3 ver.). The distribution of relaxation time (DRT) method was applied to analyze the EIS data [15].

## 3. Results and Discussion

Figure 1 shows the XRD pattern of Co-doped $Pr_{0.7}Ca_{0.3}MnO_{3-d}$ and the (112) main peak. The XRD patterns were indexed according to Caignaert et al. [16]. Each composition showed an orthorhombic structure, indicating that it was synthesized without the secondary phase formation. The Goldschmidt tolerance factor (t) of PCMCx increased from 0.908 (PCM) to 0.913 (PCMC0.3), which was predicted to have a crystal structure close to the cubic structure but showed an orthorhombic structure. Compared to PCM, PCMC0.1 showed a lower angle shift of the main peak from 32.88° to 32.84°, but as the content of

Co increased, the main peak gradually shifted to a higher angle, and PCMC0.3 showed a higher angle shift to 32.96°, indicating that the lattice contracted. The lattice contraction is caused via the substitution of Mn ions with larger atomic number Co ions: $Mn^{3+}$ (58 pm LS and 64.5 pm HS) and $Co^{3+}$ ions (54.5 pm LS and 61 pm HS) [17].

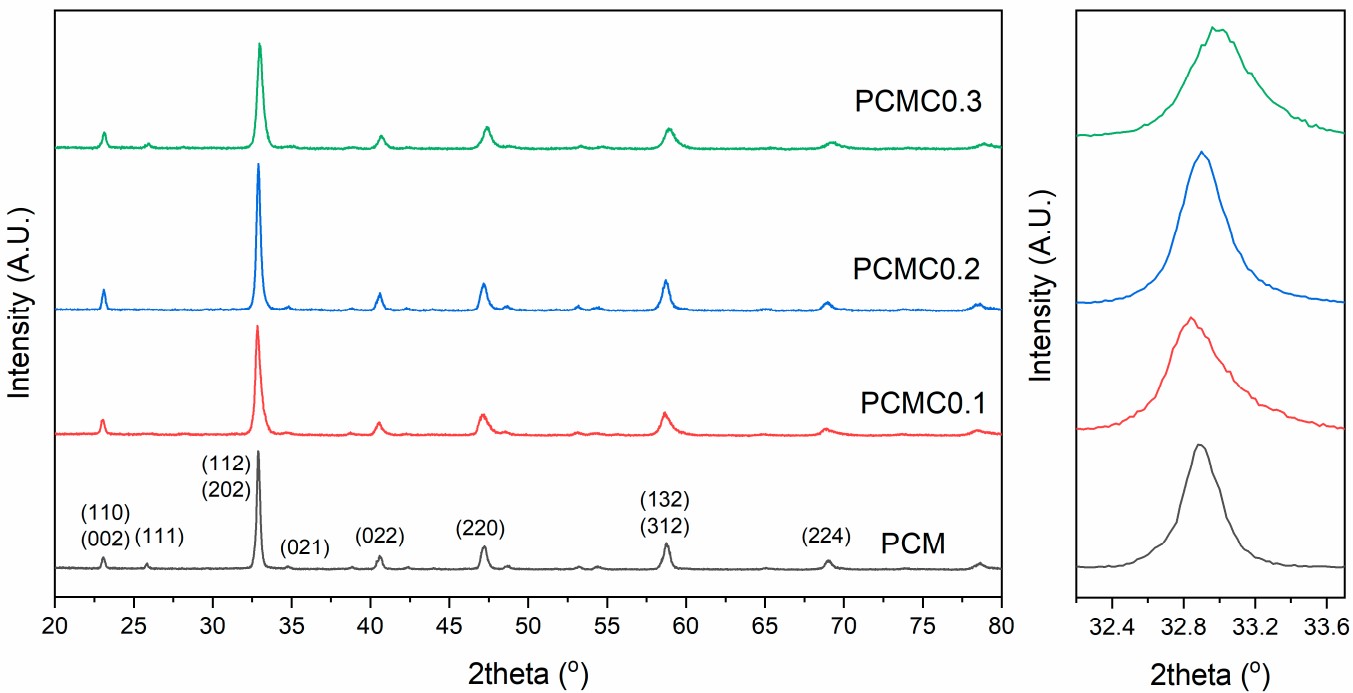

**Figure 1.** X-ray diffraction pattern and (112) peak of Co-doped $Pr_{0.7}Ca_{0.7}MnO_{3-d}$.

Figure 2 shows the core level spectra of B site metal ions and oxygen ions of Co-doped $Pr_{0.7}Ca_{0.3}MnO_{3-d}$. The core level spectra of the Co 2p1/2 states were observed at 790 eV to 810 eV and consisted of three peaks at 796.4 eV, 794.9 eV, 797.4 eV, and 802.5 eV that can be assigned to $Co^{2+}$, $Co^{3+}$, $Co^{4+}$, and satellite [18,19]. The Mn 2p2/3 spectrum consists of three peaks at 640.3, 641.7, and 643.8 eV that can be assigned to $Mn^{2+}$, $Mn^{3+}$, and $Mn^{4+}$ ions [20]. The O 1s core level spectra consist of peaks formed at 528.86 eV, 530.95 eV, and 533.4 eV, which correspond to lattice oxygen (O1), oxygen vacancy or surface absorbed oxygen (O2), and absorbed water species (O3), respectively [21]. The fitting results of the above Mn, Co, and O core level spectra are summarized in Table 1, and Pr 3d and Ca 2p also fitted according to Mekki et al. and Wan et al., respectively, and maintained oxidation values of 3+ and 2+, respectively [22,23].

The fitting results of the O 1s spectra show that the intensity of O 1s decreased and then gradually increased with Co doping, indicating the formation of oxygen vacancies in the lattice. Co, which has a similar oxidation value to Mn, was able to form oxygen vacancies, according to Lv et al. In addition, the ratio of the O2 peak increased contrary to the trend of O1, which indicates that the B-O covalency was improved, and it is known that improved covalency indicates better ORR reactivity. O3 represents an absorbed water species [22,24].

It was found that the Co ions in PCMC0.1 had oxidation values of 2+ and 3+, but as the Co content gradually increased, the ratio of $Co^{2+}$ decreased and the ratio of $Co^{3+}$ increased, which is consistent with the trend of the O1 peak observed in the O 1s spectra. In addition, the overall oxidation value of the B site ions gradually increased with the increase in the Co doping amount, which is consistent with the lattice contraction with the increase in Co content confirmed using XRD results.

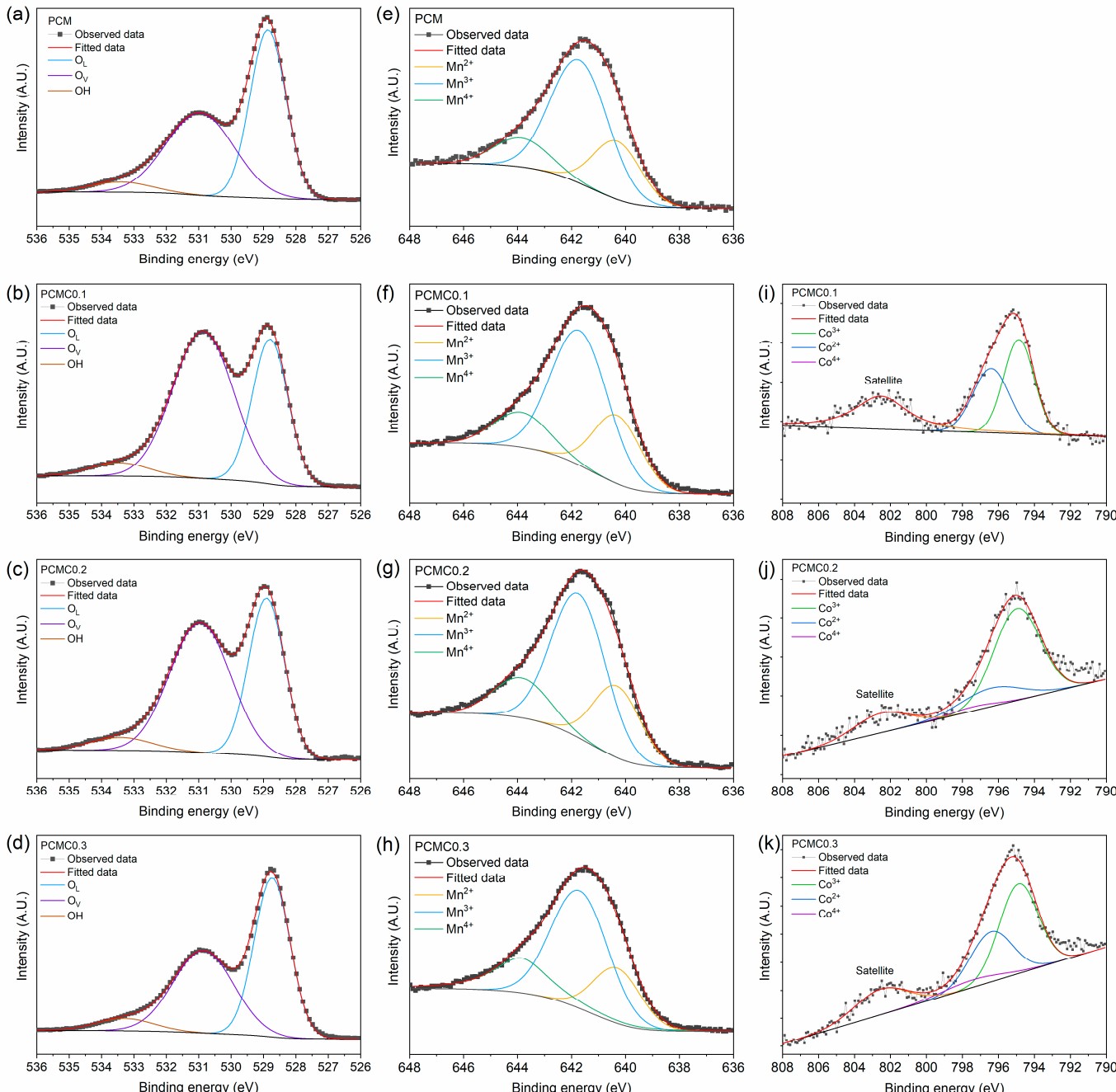

**Figure 2.** X-ray photoelectron spectra of PCMCx and peak deconvolution result: (**a–d**) O 1s core level spectra, (**e–h**) Mn 2p 3/2 core level spectra of PCM and PCMC0.1–0.3, and (**i–k**) Co 2p 1/2 core level spectra of PCMC0.1–0.3.

Figure 3 shows the electrochemical impedance spectra of each composition from 550 to 750 °C, and the ohmic resistance is subtracted to compare the polarization resistance. The polarization resistances are summarized in Table 2. PCMC0.1 exhibited the smallest polarization resistivity at all temperatures and was reduced by more than 60% compared to PCM. This is consistent with the formation of oxygen vacancies and improvement in B-O covalency, as confirmed using XPS results, and this trend is also confirmed by the increase in polarization resistance with increasing Co content.

**Table 1.** O 1s, Mn 2p 3/2, and Co 2p 1/2 XPS fitting results of PCMCx and average oxidation number.

| O 1s | O1 | O2 | O3 |
|------|------|------|------|
| PCM | 49.400% | 45.567% | 5.033% |
| PCMC0.1 | 34.805% | 60.031% | 5.165% |
| PCMC0.2 | 39.679% | 54.841% | 5.479% |
| PCMC0.3 | 49.868% | 43.843% | 6.288% |

| Mn 2p 3/2 | $Mn^{2+}$ | $Mn^{3+}$ | $Mn^{4+}$ | Average Oxidation Number |
|-----------|-----------|-----------|-----------|--------------------------|
| PCM | 24.142% | 61.829% | 14.029% | 2.899 |
| PCMC0.1 | 25.265% | 59.626% | 15.109% | 2.898 |
| PCMC0.2 | 25.295% | 58.368% | 16.337% | 2.910 |
| PCMC0.3 | 21.958% | 55.977% | 22.065% | 3.001 |

| Co 2p 1/2 | $Co^{2+}$ | $Co^{3+}$ | $Co^{4+}$ | Average Oxidation Number |
|-----------|-----------|-----------|-----------|--------------------------|
| PCM | - | - | - | - |
| PCMC0.1 | 44.894% | 55.106% | 0.000% | 3.165 |
| PCMC0.2 | 35.004% | 57.447% | 7.549% | 3.275 |
| PCMC0.3 | 19.789% | 76.885% | 3.326% | 3.449 |

Average oxidation state of A site metal: Pr = 3+, Ca = 2+.

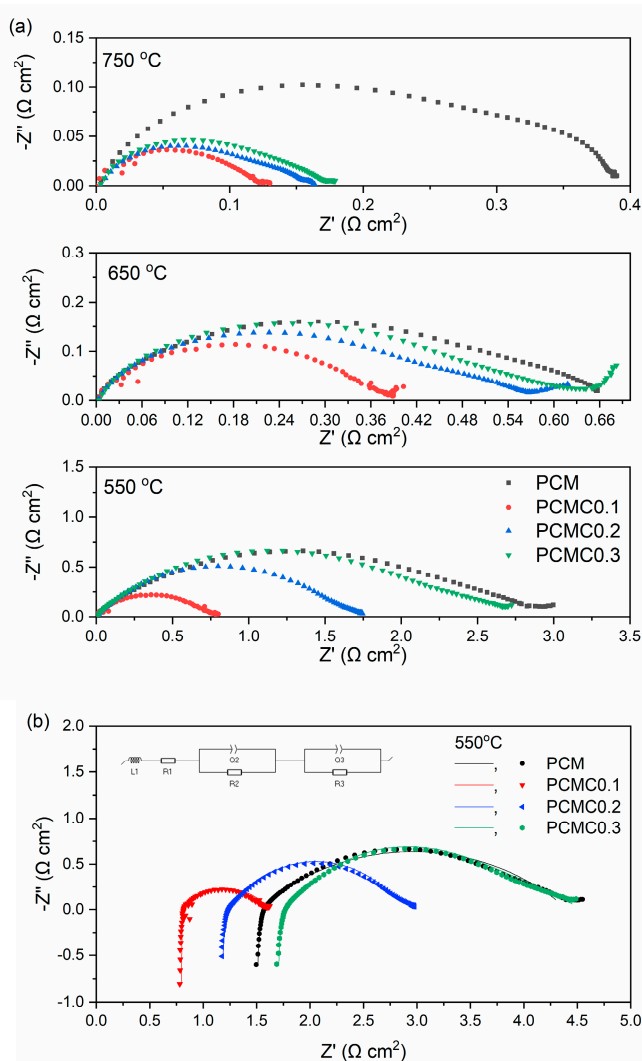

**Figure 3.** (**a**) Electrochemical impedance spectra of PCMCx. Rohm was subtracted to emphasize the polarization resistance and (**b**) fitting result at 550 °C.

**Table 2.** Ohmic resistance ($R_{ohm}$) and polarization resistance ($R_{pol}$)of PCMCx via temperature.

|  |  | **550 °C** | **650 °C** | **750 °C** |
|---|---|---|---|---|
| PCM | $R_{ohm}$ | 1.564 Ω cm² | 0.597 Ω cm² | 0.366 Ω cm² |
|  | $R_{pol}$ | 2.997 Ω cm² | 0.655 Ω cm² | 0.390 Ω cm² |
| PCMC0.1 | $R_{ohm}$ | 0.823 Ω cm² | 0.539 Ω cm² | 0.320 Ω cm² |
|  | $R_{pol}$ | 0.787 Ω cm² | 0.38 Ω cm² | 0.127 Ω cm² |
| PCMC0.2 | $R_{ohm}$ | 1.230 Ω cm² | 0.500 Ω cm² | 0.312 Ω cm² |
|  | $R_{pol}$ | 1.746 Ω cm² | 0.567 Ω cm² | 0.161 Ω cm² |
| PCMC0.3 | $R_{ohm}$ | 1.766 Ω cm² | 0.814 Ω cm² | 0.461 Ω cm² |
|  | $R_{pol}$ | 2.670 Ω cm² | 0.682 Ω cm² | 0.179 Ω cm² |

The electrochemical kinetics of PCMCx using DRT analysis is shown in Figure 4. The electrochemical process appears as a peak in the DRT spectrum, and the impedance data of the oxygen-reduction reaction can be used to analyze the main processes involved in the electrode reaction. In Figure 4, the DRT spectrum can be divided into a high-frequency (HF) region, which is found at frequencies above $10^3$ Hz, an intermediate frequency (IF) region between $10^3$ and $10^0$ Hz, and a low-frequency (LF) region below $10^0$ Hz, with each region corresponding to a separate electrochemical reaction. The characteristic frequency of a reaction is inversely proportional to the relaxation time of the electrode reaction; therefore, the higher the characteristic frequency, the faster the relaxation rate of the electrode process. In addition, the integral area of the peak represents the resistance of the corresponding reaction [25,26].

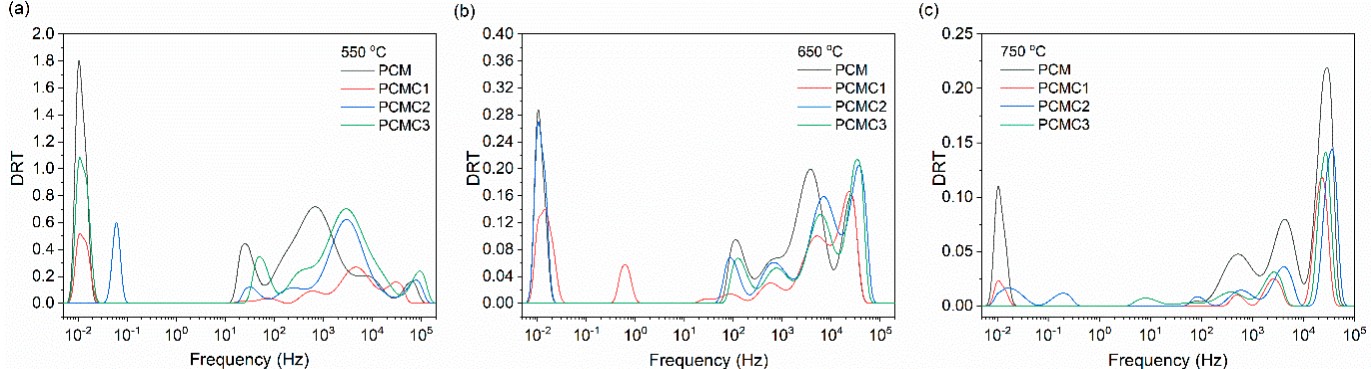

**Figure 4.** Distribution of relaxation time spectra of PCMCx via temperature: (**a**) 550 °C, (**b**) 650 °C, and (**c**) 750 °C.

The LF region is the region for the reaction in which oxygen molecules are diffused and adsorbed onto the electrode, and at 550 °C, PCM exhibited the largest LF resistance, which rapidly decreases with increasing temperature. The processes contributing to the HF region are likely related to one or more charge-transfer processes across the interface, such as the transfer of oxygen ions ($O^{2-}$) at the TPB/electrode–electrolyte interface. This region is more dominant at higher temperatures than at lower temperatures, as evidenced by the particularly high peak at 750 °C.

Electrochemical reactions corresponding to the peaks in the IF region can include reactions such as adsorption/desorption of oxygen, dissociation, and surface migration, accounting for most of the polarization resistance in EIS. This region is strongly influenced by the formation of oxygen vacancies, which is confirmed by the decrease in O1 in XPS, and the enhancement in B-O covalency, which is confirmed by the O2 peak, which corresponds to the smallest O1 ratio and the largest O2 ratio in PCMC0.1. Therefore, the polarization resistance is lower than that of PCM due to the synergetic effect of oxygen vacancy formation and B-O covalency enhancement via Co doping, and as the Co content

increases again, the oxygen vacancy decreases, and the B-O covalency decreases, gradually increasing the resistance.

## 4. Conclusions

In this study, the effect of Co doping on the electrochemical properties of $PrCaMnO_{3-d}$ was analyzed from the perspective of the change in the bonding structure of oxygen. PCMCx showed a gradual lattice contraction with increasing Co content, which is believed to be because Co ions are smaller than Mn ions, and the B site ions are gradually oxidized and become smaller in size with increasing Co content. Among PCMCx, PCMC0.1 showed a lower average oxidation value of the B site than PCM, and the lattice oxygen decreased, indicating that oxygen vacancies were formed in the lattice. As the content of Co increased, the oxidation value of Mn and Co gradually increased, and the content of oxygen vacancies in the lattice decreased. In addition, the absorbed oxygen species in PCMC0.1 increased, resulting in an increase in B-O covalency. In EIS, PCMC0.1 showed the smallest polarization resistance, and DRT analysis showed that the decrease in resistance in the IF region, which represents the electrochemical reaction with oxygen, was the main cause.

**Author Contributions:** Conceptualization, H.L.; methodology, S.L.; software, K.J.; validation, K.J., S.L., and H.L.; formal analysis, S.L.; investigation, S.L.; writing—original draft preparation, K.J.; writing—review and editing, H.L.; visualization, K.J.; project administration, H.L.; funding acquisition, H.L. All authors have read and agreed to the published version of the manuscript.

**Funding:** This work was supported by the Korea Institute for Advancement of Technology (KIAT) grant funded by the Korea Government (MOTIE) (P0008335, HDR Program for Industrial Innovation).

**Institutional Review Board Statement:** Not applicable.

**Informed Consent Statement:** Not applicable.

**Data Availability Statement:** The data presented in this study are available in the article.

**Conflicts of Interest:** The authors declare no conflict of interest.

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
