# Peer review of "Oxygen-Bonding State and Oxygen-Reduction Reaction Mechanism of Pr0.7Ca0.3Mn1−xCoxO3−d (x = 0, 0.1, 0.2, 0.3)"

_ceramics, doi:10.3390/ceramics6040146_

Round 1
Reviewer 1 Report
Comments and Suggestions for Authors
Some abbreviations are used without any explanation. Despite most of them will be understandable for expert readers, a introduction of abbreviations would be better.
The naming of the materials is not uniform/consistent. In Fig. 1 and Fig. 3 the composites are named PMC0.1/0.2/0.3 instead of PMC1/2/3.
EIS:
For the EIS spectra "ohmic resistance" is substracted. I think the ohmic resistance for each sample should be give together with at least one raw spetrum. This would be helpful to evaluate the general behaviour and electrode effects.
How was the polarization resistance obtained? Just read from the spectrum? I would assume to use a fitting model. It is clear from the spectrum (and your DRT analysis) that there are several process observable. A educated fitting would help to seperate these processes and to compare the resistances.
Comments on the Quality of English Language
For the introduction and most of the manuscript the sentences are very concise, but not always precise. A little bit more context would help to read the article.
Author Response
Some abbreviations are used without any explanation. Despite most of them will be understandable for expert readers, a introduction of abbreviations would be better.
Following the comment, we have revised the full words of SOFC, LSCF, BSCF, ORR, EDTA, SDC and written the abbreviations in parentheses.
The naming of the materials is not uniform/consistent. In Fig. 1 and Fig. 3 the composites are named PMC0.1/0.2/0.3 instead of PMC1/2/3.
Thank you for your detailed comment. We have revised the draft as you suggested (PCM, PCMC0.1, PCMC0.2, PCMC0.3) and marked with blue for the revised part.
EIS:
For the EIS spectra "ohmic resistance" is substracted. I think the ohmic resistance for each sample should be give together with at least one raw spetrum. This would be helpful to evaluate the general behaviour and electrode effects.
Thank you for the informative comment. As your comment, we added the additional EIS spectra for 550℃ as figure 3 (b) to compare the ohmic resistance by composition. And the ohmic resistances were added in table 2.
How was the polarization resistance obtained? Just read from the spectrum? I would assume to use a fitting model. It is clear from the spectrum (and your DRT analysis) that there are several process observable. A educated fitting would help to seperate these processes and to compare the resistances.
Thank you for the comment. The polarization resistance was obtained from fitting result. We added the additional EIS spectra for 550℃ with the equivalent circuit as figure 3 (b).

Reviewer 2 Report
Comments and Suggestions for Authors
The article presents the results of the study of Co-doped PrCaMnO3-d regarding the change of oxygen bonding state at different doping levels. This study is expected to contribute to the development of electrochemical cells for efficient chemical energy conversion when the operating temperature is reduced to the intermediate temperature range (600~800℃).
The authors describe in detail the synthesis procedure of Pr0.7Ca0.3Mn1-xCoxO3-d powders with different compositions.
A comprehensive study of the influence of Co doping on the B-O bond structure and electrochemical properties of PCMCx was carried out based on X-ray diffraction, X-ray photoelectron spectroscopy, electrochemical impedance spectroscopy, and distribution of relaxation time methods. The methods of characterization of the studied compounds are properly chosen and described in sufficient detail.
I would like to ask the authors how they explain the increase in asymmetry and FWHM of the (112) peak on the XRD spectra of compounds PCMC0.1 and PCMC0.3.
I would recommend that the authors change the color of one of the two blue lines in Figure 2(a-d), as they look very similar. Also, the caption in Figure 2 is hard to read because of its small size.
Overall, the article made a good impression and I am happy to recommend it for publication
Author Response
The article presents the results of the study of Co-doped PrCaMnO3-d regarding the change of oxygen bonding state at different doping levels. This study is expected to contribute to the development of electrochemical cells for efficient chemical energy conversion when the operating temperature is reduced to the intermediate temperature range (600~800℃).
The authors describe in detail the synthesis procedure of Pr0.7Ca0.3Mn1-xCoxO3-d powders with different compositions.
A comprehensive study of the influence of Co doping on the B-O bond structure and electrochemical properties of PCMCx was carried out based on X-ray diffraction, X-ray photoelectron spectroscopy, electrochemical impedance spectroscopy, and distribution of relaxation time methods. The methods of characterization of the studied compounds are properly chosen and described in sufficient detail.
I would like to ask the authors how they explain the increase in asymmetry and FWHM of the (112) peak on the XRD spectra of compounds PCMC0.1 and PCMC0.3.
Thank you for the kind comments. The asymmetry in this composition would be affected by the orthorhombic structure.The Goldschmidt tolerance factors (t) of the PCMC or PCM calculated to be slightly larger than 0.9, resulting in a orthorhombic structure similar to the cubic structure. Here, the (112) and (202) planes will have almost the same Bragg angle, resulting in an asymmetry of the peaks.
The broader FWHM is caused by the lattice distortion that occurs as Co replaces Mn and is a common phenomenon when doped into various ceramic materials, including perovskite oxide, as shown in the two references below.
Figure (A) X-ray diffraction data of Sr doped KNN-50/50 (K0.5Na0.5NbO3), (B) zoomed peak at 2θ ∼ 45-46.
[Hussain, F., Khesro, A., Lu, Z., Alotaibi, N., Mohamad, A. A., Wang, G., ... & Zhou, D. (2020). Acceptor and donor dopants in potassium sodium niobate based ceramics. Frontiers in Materials, 7, 160.]
Figure (a) X-ray diffraction patterns of undoped and Fe-doped ZnO nanostructures; (b) enlarged view of high-intensity peaks shifting towards lower 2θ values.
[Kumar, S., Ahmed, F., Shaalan, N. M., Arshi, N., Dalela, S., & Chae, K. H. (2023). Influence of Fe Doping on the Electrochemical Performance of a ZnO-Nanostructure-Based Electrode for Supercapacitors. Nanomaterials, 13(15), 2222.]
I would recommend that the authors change the color of one of the two blue lines in Figure 2(a-d), as they look very similar. Also, the caption in Figure 2 is hard to read because of its small size.
Modified the line colors in figure 2 (a-d) to be more distinguishable based on comments. Modified the layout from 4x3 to 3x4 for readability.
Overall, the article made a good impression and I am happy to recommend it for publication
For the figure on this answers, please see the attachment
